# Trans and gender diverse people's experiences of healthcare access in Australia: A qualitative study in people with complex needs

**Bridget Gabrielle Haire**[1]*, **Eloise Brook**[2], **Rohanna Stoddart**[1], **Paul Simpson**[1]

**1** Kirby Institute, UNSW Sydney, Kensington, New South Wales, Australia, **2** The Gender Centre, Annandale New South Wales, Australia

☯ These authors contributed equally to this work.
* b.haire@unsw.edu.au

**Data Availability Statement:** Data cannot be shared publicly because it contains sensitive information that the study participants did not consent to have shared. Data access queries may

## Abstract

### Introduction

This study aimed to explore the experiences of healthcare access in a diverse sample of trans and gender diverse individuals with complex needs using qualitative methods. We recruited 12 individuals using trans community-based networks facilitated by the Gender Centre. Each individual participated in an in-depth, semi structured interview conducted by a peer interviewer. Interviews were analysed thematically.

### Findings

Participants had a range of complex health needs to manage, including ongoing access to gender-affirming hormones, mental health care and sexual health care. Some also had chronic diseases. Accordingly, scheduling appointments and affording the co-payments required were major preoccupations. Most participants were not in full time work, and economic hardship proved to be a major compounding factor in issues of healthcare access, impacting on the choice of clinician or practice. Other barriers to accessing health included issues within health services, such as disrespectful attitudes, misgendering, 'deadnaming' (calling the person by their previous name), displaying an excessive interest is aspects of the participants' life that were irrelevant to the consultation, and displaying ignorance of trans services such that the participants felt an obligation to educate them. In addition, participants noted how stereotyped ideas of trans people could result in inaccurate assumptions about their healthcare needs. Positive attributes of services were identified as respectful communication styles, clean, welcoming spaces, and signs that indicated professionalism, care and openness, such as relevant information pamphlets and visibility of LGBTIQ service orientation. Participants valued peer-based advice very highly, and some would act on and trust medical advice from peers above advice from medical professionals.

### Conclusion

These findings demonstrate a need for comprehensive wrap-around service provision for trans people with complex needs which includes a substantial peer-based component,

be directed to the UNSW Human Research Ethics Coordinator (contact via humanethics@unsw.edu.au or via + 61 2 9385 6222).

**Funding:** Dr Bridget Haire received seed funding from the Kirby Institute, UNSW Sydney, for this study https://kirby.unsw.edu.au.

**Competing interests:** Eloise Brook is a part time staff member at the Gender Centre, a not-for-profit organisation which provides support services, referrals and counselling for TGD people, but not medical care. This does not alter our adherence to PLOS ONE policies on sharing data and materials.

and addresses physical and mental health and social services conveniently and affordably.

## Introduction

The global disease and health burden of trans and gender diverse (TGD) people are poorly understood, and the impacts of stigma, discrimination and violence are particularly serious [1]. Discriminatory or stigmatising practices affecting TGD people can occur in all aspects of their lives, including at the hands of health service providers upon whom they are reliant on for gender affirming care [2].

While TGD people are a diverse and heterogenous population [3], they are at increased risk of experiencing a range of intersecting challenges related to discrimination and harassment, both direct and indirect [4]. This can have considerable negative impact on the social determinants of health, potentially affecting work options, housing, access to social services, and healthcare [4, 5].

Previous research on the healthcare experiences of TGD people has shown that they report both positive and negative experiences of healthcare, but that negative experiences with health professionals may increase the risk of self-harm or poor mental health [2]. Compounded experiences of stigma and discrimination have, in turn, been linked to increased reporting of depression [2, 6, 7].

Discourses about mental health issues in TGD people are complicated by the diagnostic category of 'gender dysphoria'–clinically significant distress related to their assigned sex at birth– as defined in the DSM-5 [8]. 'Gender dysphoria' is different to a mental illness or disorder, and it replaces the category of 'gender identity disorder' which appeared in the previous DSM [9]. As a diagnostic category, however, it presents TGD people with a defined set of criteria that take an implicitly binary view of gender that they need to fulfil in order to access gender affirming care.

A diagnosis of 'gender dysphoria' is a gateway to accessing gender affirming treatment, including medical, social and legal remedies. This 'gateway' diagnosis is problematic not only because it is implicitly pathologizing, but because it may render a TGD person's mental health issues invisible in contexts where accessing gender affirming treatment may take priority over full and frank discussion of mental health with a clinician [10].

There are different pathways to accessing treatment. One is the 'approval letter model' which involves obtaining an approval letter from a mental health professional and may also include being referred to an endocrinologist. This model is sometimes referred to as the 'gatekeeper model', as access to care depends upon medical professionals' assessment of the TGD person. This model is endorsed by the World Professional Association for Transgender Health's (WPATH) Standards of Care [11].

A second pathway is the 'informed consent model' (also known as 'affirmation enablement'), under which a person aged 18 or older can access gender affirming hormones through primary care without outside referrals, following a process of several extended consultations during which the risks and benefits of interventions such as hormones are comprehensively explained, together with consideration of the TGD person's goals around gender affirmation and individual and family medical history and evaluation of blood tests [12, 13]. A benefit of this model is that it gives greater weight to the expertise of the TGD person and their clinician in making decisions about health, but this assumes that both have the capacity for a nuanced discussion of gender issues and their management [12].

Thus, regardless of the pathway chosen, TGD people are highly likely to have intensive contact with health services, particularly when commencing gender affirming medical care, but also in on ongoing way to maintain it. While transition related care may involve mental health practitioners, the stresses of systemic discrimination [5] and identity erasure [4] mean TGD people frequently seek out mental health care to help with the anxiety, depression and post-traumatic stress disorder (PTSD) that is a result of the negative social and familial pressures they experience [6, 14, 15].

In this study we aimed to explore the healthcare experiences of TGD people, including both general healthcare experiences and those related to their transition or gender affirming care. We wanted to investigate the experiences of 'seldom heard voices' or 'hard to reach' TGD people, as to date, the most Australian research in TGD people has had a majority of white participants [2]. In particular, we sought participants who may face further marginalisation due to low income [5], Aboriginal or Torres Strait Islander status [16], cultural identity [17], and age [18] in order to identify barriers to care and to make recommendations for change.

## Methods

### Study design

This was a qualitative, exploratory study that used semi-structured interviews with open-ended questions as the data collection tool. The research questions it addressed were developed through a dialogue between the Kirby Institute and the Gender Centre. Significant in-kind support was provided by T150, a trans-specific health service. This flexible design was chosen in order to prioritise the voice of the participants in the research, to allow the interviewer to ask prompting or follow-up issues in response to issues raised by participants, and to ensure that participants could articulate their experiences within a framework that was not overly confined by the research team's preconceptions. Having a peer interviewer–a trans woman on the research team–was an integral aspect of the study design for three key reasons. We believed that a peer interviewer would have better access to 'hard to reach' TGD people which would assist in recruitment, that she would develop better rapport with participants resulting in richer and more nuanced data, and that research teams investigating TGD issues should where possible include TGD researchers as a matter of equity [19].

**Data collection.** The data collection instrument was developed through dialogues with the first two authors. Interviews were conducted a t a convenient place for the participants, usually at the Gender Centre or T150. During the interviews, the interviewer made an initial judgment call about whether to discuss pronouns, to avoid insulting participants clearly presenting as a particular gender. Following discussion of pronouns if appropriate, and discussion of how the participants would like to be identified in the research, participants were asked to describe a healthcare experience that had 'stuck in their mind,' for whatever reason. Participants were then asked about their first experience seeing a doctor during their transition as well as their experiences of both a good and a bad healthcare interaction. They were then asked if there was anything that they would like to say now to the healthcare provider involved in the good or bad experience. Following this, participants were asked to rate their own health, and to reflect on aspects of healthcare services that they had discussed, to describe what they did and didn't like. Participants were also asked about challenges they faced accessing healthcare, and some basic demographic questions. In some cases, the order of questions was different, where the interviewer was responding to issues raised by the participant.

No intrusive questions were asked about participants' individual transition processes–all information that was forthcoming about this was volunteered by participants without prompting.

**Recruitment.** We aimed to recruit a sample that was diverse with regard to Indigenous status, ethnicity and age, and to include men, women and non-binary people. We did not aim for equal numbers in each category.

Participants were recruited through advertisements circulated at the Gender Centre where a member of the research team–the peer interviewer–worked part time. Study advertisements were circulated purposively to particular support groups that met at the centre in order to maximise diversity. These included the young women's group, the over 40s group, the non-binary group and the F2M group. Staff at the trans health service T150 also informed clients about the study. People interested in participating contacted the interviewer directly.

All participants received a payment of $AUD100 (approximately $US72). This payment was to recognise participants' role in the generation of knowledge through this study. This payment was larger than standard incentive payments for research participation, which are usually around $AUD50, if incentives are given at all.

**Analysis.** All interviews were audio recorded and transcribed verbatim by a transcriber who had signed a confidentiality agreement.

The first two authors coded the transcripts in consultation with one another.

All transcripts were read and re-read, with notes taken about thematic areas identified in each interview. At the end of this processes, a list of key themes was compiled. This was sent to the second author for discussion and feedback. All transcripts were then coded both descriptively [20] and conceptually, using NVivo 12 software. A series of four overarching themes were then identified (see results), drawing on Braun and Clarke's reflexive thematic analysis [21, 22]. Themes were discussed in depth with the second author, and a plan made for the Discussion. All participants were assigned pseudonyms.

**Ethics.** This study has ethics approval from the UNSW Human Research Ethics Committee (HC190716) and the ACON Ethics Committee (2019/25). All participants provided written informed consent, and there were no minors in the sample.

## Results

We interviewed nine transwomen, one transman and two people who identified as non-binary. Five participants identified as indigenous: three were Aboriginal or Torres Strait Islander, one was Maori and one an Indigenous Pacific Islander (for a full demographic summary see Table 1).

We identified four major themes in the interviews: complexity of care, economic hardship, parameters of competent care, and community connection with other TGD people. The first three themes had several subthemes. For complexity of care, the subthemes were complexity of accessing multiple health providers and services, complexity in mental healthcare and complexity in accessing hormones. For economic hardship, the subthemes were impacts on affording healthcare, impacts on choice of healthcare and impacts of transition related care. For parameters of competent care, we divided it into two main subthemes, barriers and positive qualities. The final theme, community connection with other TGD people, was not subdivided.

### Complexity of care

Many of the participants in this study had a range of relatively complex health needs to manage, including ongoing access to gender-affirming hormones, mental health care and sexual health care. Some also had chronic diseases including HIV, asthma, diabetes and drug and alcohol issues. Having to navigate a range of different health care providers was deemed quite burdensome to participants, as this involved the logistics of locating a suitable provider (not

**Table 1. Demographics of participants.**

| Name* | Gender | Age | Cultural identity | Occupation | Health issues |
|---|---|---|---|---|---|
| **Evelyn** | **Woman** | **31** | **Tiwi (Torres Strait Islander)** | **Unemployed** | **Mental health (depression), smoking, currently not drinking** |
| **Anahera** | **Woman** | **44** | **Maori** | **Factory worker** | **Mental health (depression, edginess), smoking, housing insecurity, over eating (sees a dietician and is member of a support group), problems accessing hormone implant, sees podiatrist.** |
| Yasmin | Woman | 58 | Australian | Freelance writer | HIV+, mental health, viral brain infection, two instances of DVT related to oestrogen; recent orchiectomy |
| Jen | Woman | 59 | Australian | Disability pensioner | Mental health (anxiety, depression); childhood trauma; severe periodontal disease; sinuses; cataracts; glaucoma, diabetes; cyst in brain; osteoporosis. |
| Rose | Woman | 44 | Australian | Unemployed | Weight concerns; recent surgery (breasts and voice); dental issues; mental health, drug and alcohol counselling |
| Vanita | Woman | 29 | Eurasian/Australian | Entertainer | Mental health, Podiatrist, sexual health |
| Raven | Non conforming | 23 | Aboriginal and Irish | Sex work and make-up | PrEP, mental health, dental, smoking |
| Ligaya | Woman | 32 | Philippina/Australian | Office manager and hairdresser | HIV, was self-medicating with hormones for 5 years |
| Taula | Woman | 39 | Indigenous Pacific Islander | Student | Appendicitis, PrEP, smoking, drinking, asylum seeker |
| Luke | Man | 25 | Caucasian | Unemployed | Shoulder injury, drug use, mental health (including anxiety/depression); asthma, allergies, overweight, polycystic ovarian syndrome, side effects from SSRI including no libido; in a study on testosterone and skin |
| Jian | Non-binary | 26 | Chinese | Unemployed | Mental health (including suicidal ideation and bi-polar disorder, possible obsessive compulsive disorder and/or attention deficit hyperactivity disorder) |
| Kirra | Woman | 19 | Aboriginal | Adult industry | Tongue tied, refused oestrogen at 17, mental health (depression), asthma |

*all names are pseudonyms.

straightforward), getting a referral, scheduling an appointment (which could take considerable time), having the means to pay if necessary, then travelling to the appointment.

**Accessing multiple health providers and services.** The frustration involved in navigating the multitude of required healthcare services and providers is expressed in these participant accounts.

> The wild goose chase is like you've gotta find a psychiatrist, you've gotta find so many health professionals. It's just . . . If you could simplify it and just go to a, a normal GP, that would be much simpler. . . If only we didn't have all these different professionals, . . . and have to co-ordinate all that stuff, then it would be much easier. -Raven

Attending a range of services was common in these participants, and some expressed that managing this complexity caused considerable stress.

> A psychiatrist, doctor, dermatologist, clinical counsellor. What's the other one? Podiatrist. Trainer. . . Like a fitness trainer. But, yeah, it's just hard 'cause I can't keep on top of things. Anahera

For Ligaya however, receiving an HIV diagnosis–something that might be expected to further complicate healthcare–helped her to prioritise and improve her health:

> I got a result that I didn't like. It turned out positive in HIV and I was so shocked. And that made me, I said to myself, "How will I turn this into a positive?" So that kind of . . . I'm very

lucky that I have family and friends supporting me and also I mean so actually I feel more healthier right now than before. Ligaya

Ligaya's paradoxical account of how her HIV diagnosis improved her health may be indicative of the strong models of HIV care in Sydney [23], such that an HIV diagnosis may have opened a clear and relatively holistic health pathway for her. These accounts thus indicate the importance of service accessibility in patient wellbeing and may recommend more centralised models in TGD healthcare provision.

Furthermore, these accounts demonstrate that trans or gender diverse status is just one aspect of health for these participants, and how their healthcare is organised around a range of different issues.

**Mental healthcare.** Almost all participants were, or had been, engaged in mental health care but many reported issues with difficulty finding an appropriate provider, having to wait to get an appointment, and concerns about how hormones may impact on mental health (issues with cost will be addressed under the subtheme 'Affording healthcare').

If I was wanting to see an extra counsellor or something like that, it can be hard to find someone that's trans-friendly. So, and, when you do declare that you are trans, they will, you know, make, they will say that, you know, "I don't have any experience in that side of things," so they make that clear with you. . .So then you have to make the decision of whether . . . It depends on what you're actually wanting counselling for, whether it's mental health or for sort of trans sort of stuff as well, you know. Rose

The process of getting in to see a mental healthcare provider was arduous for many participants.

Yeah, still trying to see a psychiatrist. I've gotta go now to some, I've been referred from the Gender Centre to go to a doctor who then can give me a referral to see a psychiatrist at their centre, maybe, but then that's, that's still weeks. Luke

Dealing with the side effects of psychiatric medication could also complicate health.

I see a psychologist and stuff but I'm trying to see a psychiatrist—yeah, psychiatrist—'cause I want to change my meds. I'm on fluoxetine [Prozac] and I wanna get off that 'cause I don't enjoy having absolutely no sex drive and I don't enjoy like the anxiety and stuff that still comes along with it. And the OCD things and stuff that the doctors think that maybe I'm Attention Deficit Disorder, but I still need to get it diagnosed and see doctors about it. Luke

On the issue of side effects, one of the participants asked whether there could be a possible link between some gender affirming hormone regimens and mental health issues.

There needs to be a lot more research on like hormones and, and like what they do to the brain, and like if they're, if it's really good to be taking as much as we're taking sometimes. Like is it good for our mental health? Because like I feel like as well my mental health has been the worst it's ever been since I started my transition. I've been happier but then I've also just felt like I've been like very unstable, like my emotions are just like up and down? Vanita

All but one of the participants in this study were receiving some form of mental health care (Taula was the exception), and the accounts here show that accessing acceptable mental health care services is not straightforward.

**Complexity in accessing hormones.** Accessing gender affirming hormones was another issue of complexity for participants, and healthcare providers were often seen as gatekeepers who prevented or complicated access.

> Why are there so many gatekeepers when it comes to health and accessing healthcare for transgenders? Like should be all across the board, you know. . . Like access[ing] hormones, gotta do this, this, this You know, it's exhausting. Rose

The current Australian system requires participants to jump through hoops, and is neither user friendly, nor makes allowances for those who are unable to follow its rules. When Kirra tried to access feminising hormones at 17 years of age in Queensland, she was only prescribed androgen blockers and not oestrogen.

> So, when I went [to see the doctor], I was like, "Hey, my friend that was in my class told me to come here." And she's like, "Yeah." . . .And then she's like, "Well, before I could do this, before I could give you hormones, I need you to go to your normal GP and get a referral. And then you can come back, and I'll prescribe the hormones." So I did that, came back. She's like, "Oh, well, you're only 17 so I'll put you on blockers." Kirra

While people younger than 18 can access gender affirming hormones in Australia with parental consent, Kirra was told that she could not as she didn't have a parent to provide that consent for her.

> And then she was like, "Well, you don't have, you don't have anybody. Like you don't have a parent or parents, so, you could either take it to court or you wait until you're 18." Kirra

Issues of hormone access were compounded for those had begun transitioning outside Australia. Several of these participants had been accessing hormones informally without medical oversight and had to switch to a different model in Australia.

> The first time when I arrived here in Australia I really wanted to continue on with my hormones. And I don't really know how it process and how I can access to a doctor, to see a doctor or something, in order for me to continue on with my hormones. . .And I was sort of like just booked in to see the doctor, just to prescribe my tablets, my hormone tablets. . .but I wanted to take it more dose than the usual one that I usually take in Tonga. So it doesn't really work out because the doctor needs to take fully check-up and test, and like the blood test and everything. Taula

Two of the transwomen participants took oestrogen alone, without an antiandrogen, in one case because she had naturally very low testosterone (Rose) and in the other, because she believed an antiandrogen would reduce her libido.

> Don't ever think of taking the blocker 'cause it's, you can't, it comes to a level you don't really feel anything. It's just like you are more like a toy. Taula

For Jian, who identifies as non-binary, a request to be put on half the usual dose of testosterone was refused by their doctor. Jian was living at home and did not want hormone related changes to be perceived by their parents.

I said I wanted to be on half dose until like sort of further notice but I think she just, she put me on full dose without telling me and hormones are quite serious things—they're not to be messed with—so I felt, I guess I felt angry that she didn't communicate that to me. The lack of communication there just sort of, instead of listening to me about what I need, she made her own mind up about what was best for me or what she thought was best for me. Jian

These accounts of hormone access demonstrate that that there are major issues for these participants about accessing optimal safe, acceptable and effective regimens for gender affirming hormones, and that there is not a great deal of trust that clinicians will listen to their patients and get it right.

## Economic hardship

The theme of economic hardship was prevalent in all interviews, and in general the greater the hardship, the greater the impact on accessing health services of choice. Most participants were not in full time work, and while Australia provides basic universal primary care, this is only cost-free in specific clinics that do not add in an additional service fee. (When services are provided free this is called 'bulk billing'–some clinicians opt to 'bulk bill' economically vulnerable patients only, while some do not provide this option at all.) Public sexual health clinics also provide free services to particular client groups that include trans and gender diverse people [24], however free or low-cost mental health care or dental health care services are relatively scarce. For our TGD participants, economic hardship proved to be a major compounding factor in issues of healthcare access. In particular, in affording healthcare and on the choice of clinician or practice.

**Affording healthcare.**   The struggle to afford to pay for mental healthcare services was raised by almost all participants. For Jen, a disability pensioner with complex care needs, planning her budget to ensure that she could access her preferred mental health service meant budgeting to pay for private health insurance (with private health insurance in Australia you have greater choice of provider). Between paying for this, and travelling to appointments, she had little money for basic needs.

Specialists don't give a break so, if I go to a specialist, there goes my two weeks. Sixty bucks of food for two weeks. So I live on leftovers from my mum in the old folks' home. She doesn't eat much. She saves me fruit. I eat what she doesn't eat. . . I try not to go near a shop. I try to not eat. But I know I need to. Jen

Many other participants discussed the cost of seeing specialist mental health providers, and some also raised the issue of co-payments for prescribed medication–particularly medication that is not publicly subsidised–which caused financial and logistical stress.

I do find a little bit with my psychiatrist, he's sort of got me on a medication with a bipolar . . . you know, I've gotta pay full price. But, you know, what do you do sort of for those things, you know? It's 20 bucks instead of six dollars, so it's one thing but . . . And there is one other medication that I take, that he won't give me PBS [subsidised prescription] but the doctor will. Jian

In addition to mental health services, the prohibitive cost of dental care was spoken of by some participants (Raven and Jen). For Jen, not being able to afford regular dental care meant that she developed severe periodontal disease.

And then dental stuff: I lost all my teeth at the bottom and then they put in, someone gave me money to get an implant, dentures, so that it screwed down. And then they went to check a few years ago and it was so infected. . .and they said, "Unless you've got lots of money, one, we've gotta cut out all the infection and cut the teeth out and, unless you've got lots of money for bone grafts and stuff".

This demonstrates how over time, being unable to afford basic self-care can cause serious and sometimes overwhelming health consequences.

**Economic hardship impacting on choice of clinician or practice.** When asked about how they choose which services or providers to attend, several participants said that the key criterion was not having to pay.

Bulk-billing and close. That's just, if it's bulk-billing, closer, I'll go check it out. Luke

Ligaya, Kirra and Taula also talked about the need for services to be close to where they were staying, and close to train lines.

While convenient and inexpensive services are obviously attractive for most healthcare users, for some of the participants in this study choice was limited by an inability to travel longer distances or to attend non-bulk billing services in order to see a preferred provider.

**Economic hardship as a barrier to transition-related care.** Several participants talked about the cost of transition related healthcare, including hormones, surgeries and cosmetic procedures to facilitate gender affirmation.

For Jian, this has meant at times needing to choose between transition related procedures and mental health care.

Finance is a huge thing because, unfortunately, a lot of specialists are very expensive and don't really fall under Medicare so much, like bulk-billing, which is really unfortunate because a lot of trans people do, indeed, need money or, or are on the poor end. I mean it costs, it costs so much to have procedures done and time. I mean I do have the time but, but the thing is time does come at a cost because the longer my mental health is poor, the sort of harder it is to I guess get out of the headspace. So, while people might say like be patient, it's not, it's not as easy as they might think. Jian

For Jen, the cost of accessing medical care for transition purposes in addition to mental and other physical healthcare means that she does not attempt any transition-related treatment.

I can't afford the medical care I need, I struggle with all of that, and finances, and all of that. So I'm really doing, my transitioning on the medical and psychiatric level is very much around my psyche and soul. Jen

Money was the key barrier identified for transition-related care by Taula.

Mainly that that's the major, major problem that we face nowadays within our trans communities. It's just financial support of us to be able to go on like with our transitions and make it happen. Taula

Vanita also noted wryly that there was sometimes direct economic discrimination against transwomen seeking to access gender affirming cosmetic procedures–while hairdressers always charged transwomen the fees for women–which tend to be higher, despite such

discrimination in pricing structures being illegal–hair removal clinics try to charge trans-women the (higher) price for men's hair removal.

## Parameters of competent care

**Barriers.**   Participants listed a range of behaviours from healthcare services that would signal that they were not suitable or competent to provide an acceptable level of service. This included disrespectful attitudes, misgendering, 'deadnaming' (calling the person by their previous name), displaying an excessive interest is aspects of the participants' life that were irrelevant to the consultation, and displaying ignorance of trans services such that the participants felt an obligation to educate them. In addition, participants noted how ste-reotyped ideas of trans people could result in inaccurate assumptions about their health-care needs.

Misgendering could occur in situations where the participant's gender was clearly stated in the paperwork.

> I went in [to the clinic] twice, specifically for my bloodwork for my transition, and like all the paperwork said for transition, like refer to me as 'he', and had my name luckily in as well as actually . . . But the blood lady still referred to me as 'she' . . .And I'm like, "It's liter-ally on the paperwork". Luke

Several participants said that they avoided clinicians who were unduly curious about them and asked a range of irrelevant or invasive questions.

> I'm really sick of being asked what my sexuality is by, like by professionals 'cause it's not relevant. . .. The thing is a lot of health professionals are really fixated on how people label themselves. That's not important. The important thing is what does the person need. Regardless of how I identify, I need testosterone. Jian

These participants who noted this excessive questioning associated it with the clinician positioning them as a curiosity rather than as a person deserving respect, and a patient needing care.

> I would try to like avoid the one who just didn't consider me female, the one that tried to ask me too many invasive questions about my transition or what everyone thinks about my like life and everything. Vanita

Rude or dismissive reception staff and waiting rooms that felt unwelcoming were also seen as barriers, whereas seeing other trans people and LGBTIQ resources in the waiting room were positive signs.

Both Ligaya and Yasmin reported instances where they perceived that healthcare providers' stereotypical idea about trans people directly and inappropriately affected the care they were offered. For Ligaya, this happened when she presented with a sore throat and an STI test was ordered without further discussion.

> It's distressing. She went on straight to give me an STI exam and, yeah, straight, like I just said I had a sore throat. Ligaya

For Yasmin, who suffered a severe collapse with a sudden onset, her illness was assumed to be drug related, despite her repeated assurance that she did not take drugs.

I got the usual, "Well, trans, living in public housing, you've either gotta be a hooker or a drug user. . . I was just trying to tell them I'm not taking drugs, I haven't taken drugs, I don't do drugs. But it's just like I was talking to a brick wall.] And they asked me about these other things to do with drug . . . how you take them, whatever, and I don't even know what that is. Yasmin

Yasmin's symptoms were in fact caused by an infection, for which she required neurological care, but she reports that the presumption they were drug induced left her 'dumped in a corner on a drip' in Accident and Emergency.

In addition to having to face stereotyped ideas of trans people from some healthcare providers, some participants reported feeling that they needed to provide education to clinicians who appeared out of their depth.

Raven, for example, described explaining the 'informed consent' model of trans-related care to a clinician, and providing her with resources to help her refer trans people appropriately to social and community services. She ascribed these gaps in knowledge to:

Transphobia, and I guess like the lack of education around trans health. . .I think there was like a bit of a hole with education because I seem to be educating doctors . . . wherever I go. Raven

Raven's quote demonstrates both her generosity–she wanted to improve services for other trans people–but also her frustration at having to take on this task, which was clearly not her responsibility.

**Positive qualities.**   In discussion of the positive qualities of a clinician or health service that participants would trust, they spoke generally of respectful communication styles, clean, welcoming spaces, and signs that indicated professionalism, care and openness, such as relevant information pamphlets and some level of visibility of LGBTIQ service orientation.

I think it's very much, the hospitality, it's, it's in a, a level of, you know, like you feel welcome. Not only that but you feel accepted in the environment. . .When you walk in, you can easily feel that people in the reception area they're smiling faces. They, you can easily tell that they're in the LGBTQI communities. . . And the way you consult or like talking with your doctor it's, it's more like, it's just more like you're talking to your friend, you know. It's very open. And, and it's easy to have the conversation of who you are, "This is what I want". Taula

For Raven it was also important that there were sign and symbols present that indicated cultural competence with Aboriginal peoples.

If I don't see like a sign or an Aboriginal flag, or something referring to Aboriginal health services, I kind of get worried 'cause I just think about the demographic of the area. Raven

As many participants reported being misgendered or deadnamed at some point in the processes of accessing care, the importance of respect–specifically, feeling respected by the healthcare provider and staff, was deemed highly important.

It's probably one of the biggest things when everyone says, whether it's transgender or anything, is respect. Treat people as equals. Don't judge. Don't shame. 'Cause even if I have to take something and I struggle to be regular, telling me I should [be better at adhering to

medical advice] doing it out of fear and guilt is not getting me to do it. But a longer journey will get me to where I need to do it if I'm going to do it. Jen

A couple of participants had formed strong relationship with individual clinicians with whom they had established good communication and trust, and then continued to see in an ongoing way. Luke talked about following a particular doctor to several different medical centres due to the quality of her care.

She's just so lovely and she's just always on the ball, and like can help you even when you know you don't, like even when you don't even know that you need help. And like she's the one that's provided me with all the information on getting my surgeries and all the name-changing, all the legal documents and all that. She's the one that provided me with all that information without even having to ask her. So I don't know; she was just very ahead of the game. Luke

For Jen, a similarly strong relationship meant that when Jen thought she could no longer attend the practice as it had ceased bulk billing, the clinician simply continued to bulk bill her, knowing that without this Jen would be unable to attend.

She never said it. She just did it. Jen

Many participants spoke warmly about services provided by T150, a trans-specific health-care service that operates one day a week, and the Gender Centre, a not-for-profit community centre that provides support, counselling, social services and referrals for TGD people at no cost. While these services were highly valued, and some participants were able to have all their health needs met by attending these services, limited hours for T150 and sometimes long waiting periods for some Gender Centre services meant that for those with complex needs, who lived out of the area, and/or had complex schedules to manage, access to other medical services was also required.

## Community connection with other TGD people

**Provision of healthcare advice.**    Most of the participants in this study talked about the importance of the trans community in terms of getting supportive healthcare information and advice. This advice included which services to attend and what to ask for.

So, I seek out advice and referral from some other transgender that have been here before me, and they refer me. Yasmin

Both Taula and Kirra first learned about gender affirming hormones from trans peers, and Ligaya, who had a bad experience with a healthcare provider, self-medicated for several years relying on the advice of trans friends to keep her safe, as did Taula and Yasmin.

Most of the transitioning that I have done is through word-of-mouth, of friends. . . 'cause my friends are trans as well. Ligaya

Taula explained that her peer group of trans friends were a critical source of what to do and what not to do. In particular, she said they provided advice on hormones and what to ask for from doctors.

If you just want to take hormone[s], just go straight to the doctor and say like you're want-ing to take hormone[s], and, "I want this and I want that." And it might be that doctor will refer you to some, some other expert but you don't have to go like in this long process of like before you're taking hormone[s]. Taula

Several participants spoke finding community with other TGD people when they moved to Sydney, Australia's largest city, from rural or remote areas, and that this was facilitated by ser-vices including the Gender Centre and T150.

Ever since I came here, it feels like, you know, like I'm just connected to everyone. . .It feels very important, you know. Like I feel supported, you know. You know, I've got help and like people who are like me. Yeah. And just like brother boys*, sisters girls**; I meet heaps of them. . ..I like most of the Gender Centre because I get more help. That's what I like about and T150 as well. Evelyn

*Aboriginal and Torres Strait Islander trans men.

**Aboriginal and Torres Strait Islander trans women.

Community connection was discussed as a critical enabler of accessing good healthcare by many participants. This connection was not available to all participants, however.

So, another one around health and wellbeing is, okay, so one of the things is around loneli-ness. Having sickness and having functional poverty. . .I can't go and do things with other people, I know I'm not well, I don't have the money, and so eventually people don't ask. And I hear about people that, that were friends and they've all gone out for someone's birth-day party, and it's like they didn't invite me. . .And loneliness is a killer. Jen

For some, economic stress can disrupt community connection, making it harder to see friends and creating further isolation, and disconnecting channels of healthcare advice.

## Discussion

This study provides unique insights into the healthcare issues faced by a sample of trans people who are marginalised and/or experience complex health needs. The results demonstrate how the challenges of gender transitioning are magnified by comorbidities including mental health issues and marginalising factors such as economic insecurity.

For these participants, the lack of regular, reliable incomes increased the hurdles that they faced to staying well, dealing with mental and/or illness and/or disability, and transitioning. The most striking finding was how lack of money and the cost of attending health services (including travel and any copayments for services and medication) constrained their choices, and left several making decisions about care on the basis of bulk billing and location rather than being able to seek out trans-friendly recommended providers. This finding is not novel, in that aligns with previous research predominantly in the United States [25, 26], but it is sig-nificant to note that it occurs even with Australia's model of primary care provision, where at least some primary care services are available free. It is also sobering to see how for one partici-pant, maintaining private health insurance in order to access her preferred mental health care provider could mean food insecurity.

Kirra's experience of being unable to access feminising hormone until aged 18 reflects cur-rent Australian law, where access for TGD people under 18 depends upon consent from a par-ent or guardian (and where, as the recent Re: Imogen case has shown, if only one of two

parents give consent, access will not go ahead) [27]. Her doctor however could have explained this limitation initially, and could have better explained that there was an option for Kirra to pursue access through the Family Court, and how she could access legal support for this. While it is arguably unlikely that Kirra would have taken this option, as a young Aboriginal person she would have been able to access free legal support. Cruz has argued that being visibly gender ambiguous can be a negative shaping factor of TGD people's social experiences and realities. In Kirra's case, earlier access to feminising hormones could have forestalled some of the discrimination and harassment she faced as a visibly gender non-conforming person by making her more visibly congruent with normative femininity [28]. While the gap between being 17 and 18 may seem insignificant period to wait for hormone treatment to a health care provider, to the young person already vulnerable to marginalisation due to her Aboriginality, having an appearance that signalled gender nonconformity may have unnecessarily exposed her to further discrimination and harassment.

Visibility is also a factor in the account provided by Jian, the non-binary person in the study, but in a different way. Jian's experience of being put on the standard dose of hormones despite their specific request for a lower dose echoes Bauer et al.'s theory of erasure, in this case erasure of non-binary identity, or collapsing their identity into that of trans man [4]. This demonstrates a lack of care–or lack of understanding–both for Jian's goals regarding their transition, and for their social circumstances, where too-visible changes may put them at risk, due to living with their parents. This example aligns with recommendation from other studies both in Australia and overseas regarding the need for increased education of medical practitioners with respect to engaging with gender diverse clients [2, 29, 30].

The importance of services sending strong visual message about who was welcome at their service–such as Aboriginal flags, mentioned by Raven, and LGBTIQ+ health information pamphlets, mentioned by Taula–echoes research by Koh, Kang and Underwood, who found that visual symbols of respect were important to gender and sexuality minorities when seeking health services [3].

Respectful communication from all staff in a service, including correct use of pronouns and of preferred names, was a key factor in allaying the anxieties that many of these participants felt when attending a health service. The incidences of misgendering and other disrespectful communication from health service providers reported here aligns with previous research [31–33]. In a US-based qualitative study, Roller et al. found that the main psychosocial problem for their TGD participants was the struggle to find TGD-sensitive health care [34]. This points to a training issue–medical education designers, both at university and continuing medical education programs, need to address this systemically as a matter of urgency [26]. Simple protocols about using pronouns for TGD people abound on the Internet–but the problem is getting clinicians and their staff to recognise the importance of getting this right, and how to ask respectfully if they are unsure [35].

Participants in this study placed a high value on TGD community connection not just for social support but for medical advice on transitioning, sometimes to the exclusion of formal medical care. While there are obvious dangers involved in self-medicating, the importance of peer voices in these participants' lives, and the fact that peers were trusted to listen to and respect participants' gender goals and health values, provides a lesson for clinicians and health services. It strongly suggests that competent TGD care should involve actively linking patients with peer networks in order to consolidate this community-based strength and counter isolation. It also suggests that clinicians should actively learn how to provide care that is truly responsive to the needs and values of the patient. For TGD people ambivalence and uncertainty that health providers will know how to meet their needs has been identified as a factor

that reinforces health disparities [36], and in our study it appears to be linked to some participants rejecting formal care.

This study has several key strengths. Working in partnership with the Gender Centre, we were able to recruit a diverse sample of TGD people with excellent representation of Indigenous peoples, who had complex care needs. Usually these participants would be 'hard to reach' [37], and they are generally underrepresented in Australian research on TGD [2, 6]. Having an 'out' trans woman as the interviewer may also have contributed to richer, more nuanced discussions of healthcare experiences.

The study also has some limitations. Recruiting through the Gender Centre networks is likely to affect findings in several ways: participants familiar with the Gender Centre may be generally better networked with trans services than other TGD with complex needs; participants may be more likely to have good access to the inner city than other TGD with complex needs; and participants with a prior relationship with the interviewer may have tailored their responses for social desirability. In addition, there are significant more transwomen in the sample, with only one trans man and one non-binary person, so issues relating to these particular genders are underrepresented in this study.

## Conclusions

Highly vulnerable TGD need healthcare services that are respectful, supportive, comprehensive, and free. The findings of this study make a strong case for the provision of multidisciplinary 'wrap around' services that address the social dimensions of disadvantage as well and physical and mental health needs for TGD with complex needs. This alone is not enough, however, as given mobility issues and the vicissitudes of life, these TGD people also need to be able to confidently access primary and tertiary care as needed. This calls for cultural training for clinicians and staff to ensure that when TGD people seek healthcare, they are not further harmed by the attitudes they encounter.

## Acknowledgments

We thank the participants who generously shared their stories with us. We would also like to acknowledge John Kaldor, Raewyn Connell, Denton Callander and Teddy Cook for stimulating conversations that contributed to the conceptualisation of this research project. We would also like to thank T150 and the Gender Centre for their practical support of this study which included help with recruitment and provision of friendly and culturally appropriate places to conduct interviews.

## Author Contributions

**Conceptualization:** Bridget Gabrielle Haire, Eloise Brook, Paul Simpson.

**Data curation:** Bridget Gabrielle Haire, Eloise Brook, Rohanna Stoddart.

**Formal analysis:** Bridget Gabrielle Haire.

**Project administration:** Bridget Gabrielle Haire.

**Supervision:** Bridget Gabrielle Haire.

**Writing – original draft:** Bridget Gabrielle Haire.

**Writing – review & editing:** Eloise Brook, Rohanna Stoddart, Paul Simpson.

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
