## [Decision Letter · Decision Letter 0]

13 Nov 2020

PONE-D-20-32368

Trans and gender diverse people’s experiences of healthcare access in Australia: a qualitative study in people with complex needs

PLOS ONE

Dear Dr. Haire,

Thank you for submitting your manuscript to PLOS ONE. After careful consideration, we feel that it has merit but does not fully meet PLOS ONE’s publication criteria as it currently stands. Therefore, we invite you to submit a revised version of the manuscript that addresses the points raised during the review process.

We look forward to receiving your revised manuscript.

Kind regards,

Stefano Federici, Ph.D.

Academic Editor

PLOS ONE

Additional Editor Comments:

Although the two Reviewers suggested two conflicting judgments (major revision vs. accept), however, even the Reviewer 1 who suggested a major revision has raised objections that do not substantially affect the goodness of the work. Therefore, I suggest that the Authors take advantage of Reviewer 1’s review suggestions to increase the quality of the study and make it suitable for publication.

Journal Requirements:

"We would also like to thank T150 and the Gender Centre for their support of this study."

"Dr Bridget Haire received seed funding from the Kirby Institute, UNSW Sydney, for this study https://kirby.unsw.edu.au"

Additionally, because some of your funding information pertains to commercial funding, we ask you to provide an updated Competing Interests statement, declaring all sources of commercial funding.

In your Competing Interests statement, please confirm that your commercial funding does not alter your adherence to PLOS ONE Editorial policies and criteria by including the following statement: "This does not alter our adherence to PLOS ONE policies on sharing data and materials.” as detailed online in our guide for authors  http://journals.plos.org/plosone/s/competing-interests.  If this statement is not true and your adherence to PLOS policies on sharing data and materials is altered, please explain how.

Please include the updated Competing Interests Statement and Funding Statement in your cover letter. We will change the online submission form on your behalf.

"Eloise Brook is a part time staff member at the Gender Centre, a not-for-profit organisation which provides support services, referrals and counselling for TGD people, but not medical care."

Reviewers' comments:

Reviewer's Responses to Questions

**Comments to the Author**

1. Is the manuscript technically sound, and do the data support the conclusions?

Reviewer #1: Yes

Reviewer #2: Yes

2. Has the statistical analysis been performed appropriately and rigorously? 

Reviewer #1: N/A

Reviewer #2: N/A

3. Have the authors made all data underlying the findings in their manuscript fully available?

Reviewer #1: No

Reviewer #2: Yes

4. Is the manuscript presented in an intelligible fashion and written in standard English?

Reviewer #1: Yes

Reviewer #2: Yes

5. Review Comments to the Author

Reviewer #1: 1) I dont know that the first sentence is correct in terms of being understudied: of all the research in the field, research on health and the impact of discrimination likely constitutes the bulk of research

2) The paper switches between writing out, for example, trans woman as two words, and combining them as one. Having it uniform as two words throughout would be preferable

3) Not sure that AMAB and AFAB are necessary in the table?

4) The following chapter might be of interest to the authors given the focus on complexity of care: Riggs, D.W., & Bartholomaeus, C. (2017). The disability and diagnosis nexus: Transgender men navigating mental health care services. In C. Loeser, V. Crowley & B. Pini (Eds.) Disability and masculinities: Corporeality, pedagogy, and the critique of otherness (pp. 67-84). London: Palgrave.

5) I think actual pseudonyms would be preferable to initials. Also, the method doesnt state if these initials are pseudonyms

6) The results sit on the side of a 'extract and comment' approach, which makes for a rather thin analysis of the data. Ideally, a little more detailed analysis of each extract would be undertaken: as per Braun and Clarke, a focus on latent as well as semantic meaning. This might mean reducing the number of extracts in each sub theme as at present they are rather long

7) Some of the themes or sub themes have specific titles when the extracts are actually quite broad. The mental health sub theme, for example, includes extracts that have seemingly little to do with mental health. Either the contents of the themes needs to be clarified, or a more indicative title used. Reducing the number of extracts as per 6 might also help with this.

8) The sub theme title 'attributes' is not descriptive (ie it doesnt describe the theme contents)

Reviewer #2: This article contributes to a small but important body of knowledge about the health care needs of a hard to reach group. The authors acknowledge that it is a small sample and not representative in any sense, nevertheless the data collected is valuable in informing reform of the health system.

The study is well designed and the methodology sound. The literature is adequate for the scale of the study. The research has clearly been carried out with a great deal of cultural sensitivity and so serves to demonstrate how data can be collected from those who are disenfranchised and so often not heard.

The data is used to shape practical suggestions about health care provision in the future for those with complex needs. It makes a valuable contribution to the body of knowledge in relation to transgender people.

6. PLOS authors have the option to publish the peer review history of their article (what does this mean?). If published, this will include your full peer review and any attached files.

Reviewer #1: No

Reviewer #2: **Yes: **Emeritus Professor Anne Mitchell

---

## [Author Response · Author response to Decision Letter 0]

4 Jan 2021

Thanks too the reviewers for their thoughtful and detailed recommendations.

Reviewer #1

1) I don’t know that the first sentence is correct in terms of being understudied: of all the research in the field, research on health and the impact of discrimination likely constitutes the bulk of research

We have changed the first sentence to:

The global disease and health burden of trans and gender diverse (TGD) people are poorly understood, and the impacts of stigma, discrimination and violence are particularly serious.

2) The paper switches between writing out, for example, trans woman as two words, and combining them as one. Having it uniform as two words throughout would be preferable

Thank you for this observation – we have identified all instances of the combining into one word, and changed to two words.

3) Not sure that AMAB and AFAB are necessary in the table?

Our research team had been ambivalent about the inclusion of AMAB and AFAB, and we are very happy to take reviewer 1’s advice on this. These have now been removed (Table 1, pp8-10)

4) The following chapter might be of interest to the authors given the focus on complexity of care: Riggs, D.W., & Bartholomaeus, C. (2017). The disability and diagnosis nexus: Transgender men navigating mental health care services. In C. Loeser, V. Crowley & B. Pini (Eds.) Disability and masculinities: Corporeality, pedagogy, and the critique of otherness (pp. 67-84). London: Palgrave.

Thank you for recommending this chapter. We have included a reference to it on page 4.

 This ‘gateway’ diagnosis is problematic not only because it is implicitly pathologizing, but because it may render a TGD person’s mental health issues invisible in contexts where accessing gender affirming treatment may take priority over full and frank discussion of mental health with a clinician.(10)

We have also removed the word ‘comorbidities’ in the manuscript, in response to the critique of that term made in this chapter(see line 546).

5) I think actual pseudonyms would be preferable to initials. Also, the method doesn’t state if these initials are pseudonyms

We have changed all initials to pseudonyms and noted this in the text (lines 166 and 180).

6) The results sit on the side of a 'extract and comment' approach, which makes for a rather thin analysis of the data. Ideally, a little more detailed analysis of each extract would be undertaken: as per Braun and Clarke, a focus on latent as well as semantic meaning. This might mean reducing the number of extracts in each sub theme as at present they are rather long

We thank the reviewer for this comment and acknowledge that there is a descriptive emphasis in this study. The research team wants to keep the extracts in the results section relatively long so a sense of the voices of the participants can be discerned in the text. 

7) Some of the themes or sub themes have specific titles when the extracts are actually quite broad. The mental health sub theme, for example, includes extracts that have seemingly little to do with mental health. Either the contents of the themes needs to be clarified, or a more indicative title used. Reducing the number of extracts as per 6 might also help with this.

In the originally submitted version mental health was specifically addressed in the subtheme, Complexity in mental healthcare (lines 229-261). In response to the reviewer, we have renamed this subtheme ‘mental healthcare’. The issue of cost as a barrier to all forms of healthcare, including mental health, was then addressed under the theme ‘affording healthcare’ under the subtheme ‘economic hardship’. In this subtheme, the cost of mental healthcare is discussed alongside costs of dental healthcare and costs of prescription medication. We think that this is well justified because costs of healthcare was a significant issue for all participants, and this was not limited to mental health. Keeping this theme separate allows the reader to see the catastrophic impacts of delaying dental over time, with the example of Jen losing all her teeth and also bone mass due to her long term economic deprivation. 

8) The sub theme title 'attributes' is not descriptive (ie it doesn’t describe the theme contents)

We agree with the reviewer, and have changed the name of this subtheme to ‘positive qualities’.

Reviewer #2 

This article contributes to a small but important body of knowledge about the health care needs of a hard to reach group. The authors acknowledge that it is a small sample and not representative in any sense, nevertheless the data collected is valuable in informing reform of the health system.

The study is well designed and the methodology sound. The literature is adequate for the scale of the study. The research has clearly been carried out with a great deal of cultural sensitivity and so serves to demonstrate how data can be collected from those who are disenfranchised and so often not heard.

The data is used to shape practical suggestions about health care provision in the future for those with complex needs. It makes a valuable contribution to the body of knowledge in relation to transgender people.

We thank the reviewer for her supportive comments.

---

## [Decision Letter · Decision Letter 1]

11 Jan 2021

Trans and gender diverse people’s experiences of healthcare access in Australia: a qualitative study in people with complex needs

PONE-D-20-32368R1

Dear Dr. Haire,

We’re pleased to inform you that your manuscript has been judged scientifically suitable for publication and will be formally accepted for publication once it meets all outstanding technical requirements.

Kind regards,

Stefano Federici, Ph.D.

Academic Editor

PLOS ONE

Additional Editor Comments (optional):

Reviewers' comments:

Reviewer's Responses to Questions

**Comments to the Author**

1. If the authors have adequately addressed your comments raised in a previous round of review and you feel that this manuscript is now acceptable for publication, you may indicate that here to bypass the “Comments to the Author” section, enter your conflict of interest statement in the “Confidential to Editor” section, and submit your "Accept" recommendation.

Reviewer #1: All comments have been addressed

2. Is the manuscript technically sound, and do the data support the conclusions?

Reviewer #1: Yes

3. Has the statistical analysis been performed appropriately and rigorously? 

Reviewer #1: N/A

4. Have the authors made all data underlying the findings in their manuscript fully available?

Reviewer #1: Yes

5. Is the manuscript presented in an intelligible fashion and written in standard English?

Reviewer #1: Yes

6. Review Comments to the Author

Reviewer #1: The authors have made comprehensive changes to the manuscript, which is much improved as a result. I appreciate the points the authors made about the themes developed, and accept that there are differing ways of reporting a thematic analysis.

7. PLOS authors have the option to publish the peer review history of their article (what does this mean?). If published, this will include your full peer review and any attached files.

Reviewer #1: No

---

## [Editor Report · Acceptance letter]

14 Jan 2021

PONE-D-20-32368R1 

Trans and gender diverse people’s experiences of healthcare access in Australia: a qualitative study in people with complex needs 

Dear Dr. Haire:

I'm pleased to inform you that your manuscript has been deemed suitable for publication in PLOS ONE. Congratulations! Your manuscript is now with our production department. 

Kind regards, 

on behalf of

Prof. Stefano Federici 

Academic Editor

PLOS ONE